# Syringin and Phillygenin—Natural Compounds with a Potential Role in Preventing Lipid Deposition in Macrophages in the Context of Human Atherosclerotic Plaque

**DOI:** 10.3390/ijms26136444

**Published:** 2025-07-04

**Authors:** Agnieszka Filipek, Agnieszka Sadowska, Monika Skłodowska, Maja Muskała, Edyta Czepielewska

**Affiliations:** 1Chair and Department of Pharmaceutical Biology, Medical University of Warsaw, Banacha 1, 02-097 Warsaw, Poland; asadowska001@gmail.com (A.S.); m.sklodowska27@gmail.com (M.S.); majkamuskala@gmail.com (M.M.); 2School of Health and Medical Sciences, Vizja University, Okopowa 59, 01-043 Warsaw, Poland; edczepielewska@gmail.com

**Keywords:** ABCA1, atherosclerosis, CD36, foam cells, phillygenin, syringin

## Abstract

Syringin is a phenylpropanoid glycoside isolated from the bark of *Syringa vulgaris*. Phillygenin is a lignan obtained mainly from the fruits and flowers of *Forsythia intermedia*. Both compounds have shown potent anti-inflammatory and antioxidant properties. We investigated the potential role of syringin and phillygenin in preventing lipid deposition in macrophages. Syringin and phillygenin significantly (*p* < 0.001) reduced lipid deposition in macrophages in a dose-dependent manner. For syringin, the greatest reduction in CD36 receptor expression was found to be over 80% (50 μg/mL) compared to the cholesterol-stimulated control (*p* < 0.001). Phillygenin inhibited CD36 receptor expression by approximately 25% (50 μg/mL), compared to the stimulated control (*p* < 0.05). For syringin, the CD36 receptor regulation pathway was PPAR-γ dependent. Phillygenin showed a statistically significant (*p* < 0.001) increase in the expression of the ABCA1 transporter: 2.5-fold (10 μg/mL), 3-fold (20 μg/mL) and 4-fold (50 μg/mL) compared to the cholesterol-stimulated control. Syringin did not significantly increase ABCA1 expression. For phillygenin, the activation pathway of the ABCA1 transporter was HO-1dependent. Our study showed that syringin inhibits the cholesterol-induced differentiation of macrophages into foam cells. Moreover, phillygenin increased cholesterol efflux from macrophages. Therefore, syringin and phillygenin may be valuable agents in the prevention of early and late atherosclerosis.

## 1. Introduction

Atherosclerosis is a chronic, progressive vascular disease characterised by the appearance of specific changes in the arterial walls—atherosclerotic plaques—that can lead to cardiovascular events such as myocardial infarction, unstable angina or ischaemic stroke. Atherosclerotic cardiovascular disease (ASCVD) remains one of the leading causes of death worldwide [1]. The central cells in atherosclerosis are macrophages, which differentiate and mature from circulating monocytes. Two major macrophage subpopulations with distinct functions include classically activated or inflammatory (M1) and alternatively activated or anti-inflammatory (M2) macrophages. However, macrophages are highly plastic and can change their phenotype in response to micro-environmental changes during atherogenesis [2]. Macrophages are important regulators of plasma lipoproteins, whereby excessive lipid accumulation in macrophages induces the expression of uptake receptors such as scavenger receptors (e.g., CD36, SRA1) and downregulates the expression of efflux transporters such as ATP-binding cassette A1 and G1 (ABCA1, ABCG1), thereby promoting foam cell formation—a key step in the initiation and progression of atherosclerosis. These receptors and transporters can be regulated via different pathways, including transcriptional modulation mediated by nuclear receptors such as peroxisome proliferator-activated receptor gamma (PPAR-γ) [1]. The activation of PPAR-γ proteins induces the expression of the CD36 receptor and stimulates cholesterol accumulation in macrophages [3]. Under physiological conditions, the activation of transcription factors such as erythrocyte nuclear factor 2-related factor 2 (Nrf2) initiates the transcription of antioxidant genes, including heme oxidase 1 (HO-1), which protects against cholesterol accumulation [4]. HO-1 may induce the expression of the ABCA1 transporter, thereby promoting reverse cholesterol transport from macrophages [5]. Furthermore, HO-1 can suppress inflammation by promoting a shift in the macrophage phenotype from the inflammatory M1 to the anti-inflammatory M2 [6].

However, under pathophysiological conditions (chronic inflammation, oxidative factors), cholesterol efflux is insufficient, leading to unrestricted cholesterol accumulation in macrophages (Figure 1).

Although conventional therapies, such as statins, have shown efficacy in reducing cardiovascular risk, they focus on systemic lipid lowering rather than directly targeting foam cells. Therefore, targeted therapies that modulate signaling pathways and promote cholesterol efflux may be required to fully address the cellular pathology of atherosclerosis [7].

Syringin (Syr, Figure 2A), also known as eleutheroside B, is a naturally occurring phenylpropanoid glycoside found in a variety of plant species, including common lilac (*Syringa vulgaris*, Oleaceae) and eleutherococcus (*Eleutherococcus senticosus*, Araliaceae) [8]. It exhibits a wide range of pharmacological activities, including anti-inflammatory, antioxidant [9], neuroprotective [10] and immunomodulatory properties [11]. In the context of inflammatory diseases such as rheumatoid arthritis, syringin has been shown to suppress macrophage M1 polarization and reduce the secretion of key pro-inflammatory cytokines such as tumor necrosis factor alpha (TNF-α) and interleukin-1 beta (IL-1β) via the inhibition of phosphodiesterase 4 (PDE4) [12]. Our previous study has also shown that common lilac is a valuable source of phytoactive compounds with anti-inflammatory properties, including syringin, which significantly inhibited TNF-α production and stimulated transforming growth factor beta (TGF-β) release in monocytes/macrophages treated with lipopolysaccharide (LPS) [13]. Moreover, the anti-inflammatory and antioxidant potential of syringin through the modulation of the nuclear factor kappa-light-chain-enhancer of activated B cells/nuclear factor erythroid 2-related factor 2 (NF-κB/Nrf2) signaling pathway has been demonstrated in models of acute lung injury and ulcerative colitis [14,15].

Similarly, phillygenin (Phil, Figure 2B), a lignan compound isolated mainly from forsythia (*Forsythia intermedia*, Oleaceae), has shown potent anti-inflammatory, antioxidant and immunomodulatory activities in several disease models, including ulcerative colitis [16], lung inflammation [17] and osteoarthritis [18]. Our previous research showed that phillygenin was able to stimulate the anti-inflammatory function of macrophages by inducing TGF-β release and interleukin-10 (IL-10) receptor surface expression. It also reduced TNF-α and IL-1β production and neutrophil adhesion to endothelial cells [19]. Recent evidence has revealed that phillygenin can suppress NF-κB signaling and activate the Nrf2 pathway [20].

Considering the valuable properties of compounds found in plants of the Oleaceae family, including the results of our previous studies on oleacein, a secoiridoid isolated from common privet (*Ligustrum vulgare*), which demonstrated anti-atherosclerotic potential [6,21,22], we hypothesized that syringin and phillygenin, natural compounds with high bioactivity, may also influence the prevention of cholesterol accumulation in macrophages, thereby reducing the possibility of foam cell formation. To this end, we investigated the effect of syringin and phillygenin on the regulation of CD36 receptor and ABCA1 transporter function through signaling pathways involving PPAR-γ and heme oxygenase-1 (HO-1). Understanding the effect of both compounds on selected pathogenic mechanisms of atherosclerosis may enable the development of new strategies for the prevention of early and late atherosclerotic changes.

## 2. Results

### 2.1. Effect on Cytotoxicity

Based on our previous works [13,19], we selected concentrations of the natural compounds of 10 μg/mL, 20 μg/mL and 50 μg/mL, because these concentrations were non-toxic to cells. Moreover, higher concentrations of natural compounds (e.g., 100 μg/mL) are rarely used in scientific studies, because it is difficult to achieve such high therapeutic concentrations in the body.

After the incubation of macrophages with cholesterol (20 μg/mL) and syringin or phillygenin (10 μg/mL, 20 μg/mL, 50 μg/mL), as well as kaempferol (20 μg/mL), no cytotoxic effect on the cells was observed. The IC50 value did not exceed 20% for any of the compounds at the different concentrations tested (Appendix A). Furthermore, no toxic effect on the cells was observed when macrophages were treated with rosiglitazone at a concentration of 1 μg/mL under the same conditions (Appendix A).

### 2.2. Effect of Syringin and Phillygenin on Cholesterol Deposits in Macrophages

Cholesterol accumulation by macrophages is known to lead to foam cell formation and is a hallmark of the early stages of atherosclerosis. The incubation of macrophages with cholesterol (20 μg/mL, 24 h) resulted in the deposition of lipid deposits within the cells. Treatment with syringin or phillygenin significantly reduced lipid accumulation by cholesterol-induced macrophages (Figure 3A).

We investigated whether syringin or phillygenin increased cholesterol efflux from cholesterol-induced macrophages. Quantitative analysis showed that syringin at 20 μg/mL and 50 μg/mL increased cholesterol levels in the medium by approximately 33% (*p* < 0.05) and 59% (*p* < 0.001), respectively. Syringin at 10 μg/mL showed no statistical difference in the amount of lipids in the medium compared with the medium of cholesterol-induced macrophages (Figure 3B). In the case of phillygenin, however, the lipid level in the medium was statistically higher for all concentrations of this compound compared to cholesterol-stimulated macrophages, by more than 33% (Phil10, *p* < 0.05), 53% (Phil20, *p* < 0.001) and 69% (Phil50, *p* < 0.001), respectively. We also observed that kaempferol increased cholesterol in the medium by approximately 30% (*p* < 0.05) (Figure 3B).

### 2.3. Effect of Syringin and Phillygenin on CD36 Expression

Scientific studies confirm the role of the CD36 receptor in lipid metabolism. It has been suggested that the inhibition of CD36 receptor expression leads to a reduction in lipid accumulation in macrophages, which in turn may reduce the development of atherosclerotic plaque. We investigated the effect of both syringin and phillygenin on CD36 receptor expression. Flow cytometric analysis showed that syringin reduced CD36 receptor expression in a dose-dependent manner. At a concentration of 50 μg/mL, we observed the highest reduction, with a more than 80% reduction in CD36 receptor expression compared to the cholesterol-stimulated control (*p* < 0.001). For other syringin concentrations, the decrease in receptor expression was smaller but also statistically significant (*p* < 0.001) and amounted to approximately 60% (Syr20) and 20% (Syr10), respectively (Figure 4A,B).

In contrast to syringin, phillygenin at a concentration of 10 μg/mL and 20 μg/mL did not cause a statistically significant decrease in CD36 receptor expression. Only at the highest concentration (50 μg/mL) did phillygenin inhibit CD36 receptor activity by approximately 25% (*p* < 0.05) compared to cholesterol-stimulated cells (Figure 4A,B).

Kaempferol caused a minimal but statistically insignificant decrease in CD36 receptor expression compared to the cholesterol-stimulated control (Figure 4A,B).

### 2.4. Effect of Syringin and Phillygenin on Expression of the PPAR-γ Protein

It is known that the activation of PPAR-γ proteins induces the expression of the CD36 receptor and stimulates cholesterol accumulation by macrophages, leading to the formation of foam cells [3]. The results of the experiments showed that syringin and phillygenin significantly inhibited PPAR-γ activity (*p* < 0.001). Syringin, regardless of the concentration, caused almost the complete inhibition of PPAR-γ protein activity, from 1 to 0.02 or 0.03 ± 0.01, respectively. In the case of phillygenin at a concentration of 10 μg/mL and 20 μg/mL, PPAR-γ protein expression decreased by about 50% compared to the cholesterol-stimulated macrophage (from 1 to 0.48 ± 0,05 and 0.55 ± 0.11, respectively). The highest concentration of phillygenin (50 μg/mL) eliminated PPAR-γ activity by more than 60% (from 1 to 0.35 ± 0.02). As a positive control, kaempferol inhibited PPAR-γ protein activity by approximately 50% (from 1 to 0.48 ± 0.18) (Figure 4C).

### 2.5. Effect of Syringin and Phillygenin on CD36 Protein Expression After Preincubation with Rosiglitazone (RSG)

We used rosiglitazone (1 μg/mL) as a PPAR-γ agonist to determine whether CD36 expression is indeed dependent on the PPAR-γ signaling pathway. Macrophages were pre-incubated with cholesterol (20 μg/mL) and RSG (1 μg/mL) for 24 h and then with syringin or phillygenin (10 μg/mL, 20 μg/mL, 50 μg/mL) for another 24 h. CD36 protein expression was assessed using the WB technique. The results showed a dramatic increase in CD36 protein expression for syringin (Figure 4D) up to and above the level of Chol20+RSG-stimulated cells. No such association was observed for phillygenin and kaempferol. The intensity levels of the protein bands were similar (Figure 4D).

### 2.6. Effect of Syringin and Phillygenin on Protein Level of ABCA1 Transporter

The ABCA1 transporter is a major regulator of cellular cholesterol homeostasis. Cholesterol as a substrate activates the ABCA1 protein, which acts as a cholesterol efflux pump in the cellular lipid removal pathway. Western blot analysis showed that phillygenin caused a statistically significant 2.5-fold (10 μg/mL, *p* < 0.05), 3-fold (20 μg/mL, *p* < 0.001) and 4-fold (50 μg/mL, *p* < 0.001) increase in ABCA1 protein expression compared to the cholesterol-stimulated control, respectively. Unfortunately, no differences in ABCA1 protein expression compared to the cholesterol-stimulated cells were observed for syringin at any concentration used. In contrast, kaempferol, as a positive control, increased ABCA1 expression 2-fold compared to the cholesterol-stimulated control (*p* < 0.05; Figure 5A).

### 2.7. Effect of Syringin and Phillygenin on HO-1 Intracellular Secretion

It is known that HO-1 promotes reverse cholesterol transport from macrophages by inducing the expression of the ABCA1 transporter [5]. We examined intracellular HO-1 secretion in macrophages stimulated with cholesterol and/or without syringin or phillygenin. However, phillygenin stimulated intracellular HO-1 secretion in a concentration-dependent manner. Phillygenin at a concentration of 50 μg/mL increased HO-1 secretion by about 500% (9.03 ± 0.74 pg/g protein, *p* < 0.001) compared to macrophages stimulated with cholesterol (1.79 ± 0.07 pg/g protein). A slightly lower, about 300%, increase in intracellular HO-1 secretion was observed for phillygenin at a concentration of 20 μg/mL compared to cholesterol-stimulated cells, 5.67 ± 1.08 pg/g protein and 1.79 ± 0.07 pg/g protein, respectively (*p* < 0.001). Phillygenin, at a concentration of 10 μg/mL, increased intracellular HO-1 secretion by more than 200%, from 1.79 ± 0.07 pg/g protein to 4.12 ± 0.43 pg/g protein (*p* < 0.001). Kaempferol also stimulated intracellular HO-1 secretion by more than 250% (Figure 5B).

### 2.8. Effect of Syringin and Phillygenin on Nrf2 Protein Expression

HO-1 is known to be involved in protection against cholesterol accumulation in macrophages by regulating Nrf2 expression [4]. Western blot analysis showed that phillygenin significantly increased Nrf2 protein expression at all concentrations. The highest increase in Nrf2 expression was observed for phillygenin at 20 μg/mL and 10 μg/mL, more than 3-fold and approximately 3-fold compared to Chol20, respectively. A slightly lower increase in Nrf2 expression was observed for phillygenin at 50 μg/mL, more than 2-fold compared to cholesterol-stimulated macrophages. The same result was obtained for kaempferol (Figure 5C).

## 3. Discussion

Our results show that both syringin and phillygenin can attenuate foam cell formation by inhibiting lipid accumulation by macrophages and increasing cholesterol efflux from cells. This effect was associated with the reduced expression of the CD36 receptor and increased activity of the ATP-binding cassette transporter A1 (ABCA1) protein.

Foam cells are a major component of atherosclerotic plaques, where they actively participate in the intracellular accumulation of cholesterol. Cholesterol, properly modified low-density lipoproteins, is absorbed by macrophages with the participation of scavenger receptors such as SRA1, CD36, CD68, LOX-1. As our previous studies have shown, the CD36 receptor is most involved in lipid accumulation by macrophages [21].

However, unlike other receptors (e.g., LDL receptor), excess cholesterol in macrophages does not downregulate CD36 receptor expression. It is known that unrestricted cholesterol accumulation leads to cellular cholesterol overload and foam cell formation. This hyperlipidemic state increases oxidative stress, inflammation and pathological forms of cholesterol, including ligands for CD36 [1].

ABCA1 (ATP-binding cassette transporter A1) is a transmembrane protein also involved in cholesterol homeostasis. The main function of ABCA1 is the efflux of intracellular-free cholesterol and phospholipids from cells by active transport, and combining with apolipoproteins, mainly apolipoprotein A-I, forming new HDL particles. This step of reverse cholesterol transport promotes the removal of lipid deposits from macrophages and helps prevent atherosclerosis [23].

Our studies showed that syringin significantly reduced the expression of the CD36 receptor in a dose-dependent manner, up to approximately 80% at a concentration of 50 μg/mL, compared to cholesterol-stimulated macrophages. In the case of phillygenin, no significant differences in CD36 receptor expression were observed compared to cholesterol-stimulated macrophages. Only at a concentration of 50 μg/mL did phillygenin cause an approximately 25% decrease in CD36 receptor activity compared to cells incubated with cholesterol alone.

PPAR-γ is known to mediate the upregulation of CD36 [3]. Macrophages stimulated with cholesterol and then syringin showed a significant decrease in PPAR-γ protein activity. Moreover, preincubation with the PPAR-γ agonist rosiglitazone drastically restored the upregulation of CD36 to the level of cholesterol-stimulated macrophages. Although the treatment of macrophages with phillygenin significantly reduced PPAR-γ protein expression, preincubation with the PPAR-γ agonist rosiglitazone did not cause a significant increase in CD36 receptor expression. We also obtained similar results using kaempferol as a positive control.

The results of our studies clearly indicate that syringin, in contrast to phillygenin, can limit cholesterol accumulation in macrophages by inhibiting the expression of the CD36 receptor in a PPAR-dependent pathway. Other studies on phenylpropanoid glycosides from *Tadehagi triquetrum* have also demonstrated that these compounds can significantly reduce CD36 receptor expression [24].

Completely different results were obtained for the ABCA1 transporter. Syringin did not increase ABCA1 expression at any of the concentrations used. These results are contrary to studies on *Tadehagi triquetrum* phenylpropanoid glycosides, which showed an increase in the expression of the ATP-binding cassette transporters A1 and G1 (ABCA1 and ABCG1) [24]. On the other hand, phillygenin significantly increased the activity of ABCA1 protein in a dose-dependent manner, even 4-fold (Phil50) compared to cholesterol-stimulated macrophages. Moreover, a significant increase in endocrine HO-1 secretion and Nrf2 protein expression was also obtained only for phillygenin.

According to the literature, HO-1 induction in foam cells is known to enhance reverse cholesterol transport by activating ABC transporters [5]. Furthermore, heme oxygenase 1 suppresses inflammation by promoting a shift in the macrophage phenotype from pro-inflammatory M1 to anti-inflammatory M2. This in turn may stabilize atherosclerotic plaque [6].

The regulation of HO-1 activity depends on transcription factors, i.e., Nrf2. In physiological conditions, Nrf2 is located in the cytoplasm and is associated with the Keap1 (Kelch-like ECH-associated protein 1). The protein interaction causes the continuous degradation of Nrf2, which is ubiquitinated and targeted to the proteasome, resulting in its inactivation. Therefore, the increase in intracellular HO-1 secretion is closely related to the increase in Nrf2 expression [25].

Our studies indicate that the treatment of cholesterol-stimulated macrophages with phillygenin enhances the expression of the ABCA1 transporter and that this pathway is dependent on the up-regulation of HO-1/Nrf2. An increase in ABCA1 transporter expression has also been observed in macrophages in response to other lignans, such as arctigenin (greater burdock) [26], neolignans, such as honokiol (magnolia) [27], and flavonolignans, such as silymarin (milk thistle) [28]. Promotion of the ABCA1/HO-1/Nrf2 pathway is also characteristic of other phytochemicals. In our previous work, we showed that oleacein isolated from the leaves of *Ligustrum vulgare* (Oleaceae) may intensify the efflux of lipid deposits from human macrophages [22]. Many scientific studies confirm that phytochemicals, such as flavonoids (kaempferol, catechins—green tea) [22,29], sulfur compounds (allicin—onion, garlic) [30], isothiocyanates (wasabi) [31], polyphenols (curcumin—turmeric) [4] and lipoic acid [32] activate the ABCA1/HO-1/Nrf2 pathway. Compounds with such bioactivity may have great potential in eliminating foam cells and limiting the development of atherosclerosis.

## 4. Materials and Methods

### 4.1. Chemicals and Reagents

The THP-1 cell line was from ATCC (Manassas, VA, USA). RPMI 1640 medium (with Glutamax and HEPES), FBS (fetal bovine serum), DPBS without Ca^+2^, Mg^+2^ (Dulbecco’s Phosphate Buffered Saline) and cell culture reagents were purchased from ThermoFisher Scientific, Waltham, MA, (USA).

Phorbol 12-myristate 13-acetate (PMA), dimethyl sulfoxide (DMSO), albumin bovine serum (BSA), antibiotic antimycotic solution (penicillin, streptomycin, amphotericin B), gentamycin, cell dissociation solution non-enzymatic, Oil Red O, RIPA buffer, protease inhibitor cocktail, phosphatase inhibitor cocktail and rosiglitazone were purchased from Sigma Aldrich (St. Louis, MO, USA). Kaempferol was purchased from Serva (Heidelberg, Germany). 2- propanol pure p.a. (isopropanol), formaldehyde pure p.a. (36–38%), sodium phosphate monobasic pure p.a., and sodium phosphate dibasic anhydrous pure p.a. were from Chempur (Karlsruhe, Germany). Cholesterol–Water Soluble was obtained from Sigma-Aldrich (St. Louis, MO, USA). PE anti-human CD36 and Cell Staining Buffer were from BioLegend, USA. Primary antibodies such as ABCA1, PPAR-γ, CD36 and Nrf2 were purchased from Invitrogen (Waltham, MA, USA). Rabbit anti-β-actin polyclonal antibody was from FineTest (Wuhan, China). Goat anti-rabbit IgG (H + L) secondary antibody, HRP, was purchased from ThermoFisher Scientific (Waltham, MA, USA). The Pierce BCA Protein Assay Kit was from Thermo Scientific (Waltham, MA, USA). The QuickDetect^TM^ Total Cholesterol Human Elisa Kit and HO-1 Elisa Kit were obtained from BioVision (Zurich, Switzerland) and Gentaur (Kampenhout, Belgium).

### 4.2. Syringin and Phillygenin

Syringin (4-[(1*E*)-3-Hydroxyprop-1-en-1-yl]-2,6-dimethoxyphenyl β-D-glucopyranoside) and phillygenin (4-[(1S,3aR,4R,6aR)-4-(3,4-dimethoxyphenyl)tetrahydro-1H,3H-furo [3,4c]furan-1-yl]-2-methoxy-phenol) were isolated from *Syringa vulgaris* stem bark and *Forsythia intermedia* flowers, respectively [13]. The purity of these compounds >95% was determined by HPLC-MS and NMR.

Syringin and phillygenin were dissolved in DMSO and then in DPBS buffer at pH 7.4 to a final concentration of 10 μg/mL, 20 μg/mL and 50 μg/mL. The final concentration of DMSO did not exceed 0.01% and did not influence the performed assays.

### 4.3. Cell Culture

Human THP1 monocytes were cultured in RPMI 1640 (with Glutamax, HEPES) medium supplemented with 10% fetal bovine serum (FBS), 100 U/mL penicillin, and 100 μg/mL streptomycin. THP1 monocytes were cultured at 37 °C, 95% humidity, and 5% CO_2_, and used between passages 4 and 16. The cells were seeded in 6-well plates (1 × 10^6^ cells/well) and differentiated into macrophages by preincubation with 100 ng/mL PMA for 48 h. The cells were starved in serum-free RPMI 1640 medium for 24 h before the experiments were performed. Then, THP1 macrophages were incubated with vehicle (equal amount of culture medium) and cholesterol (20 μg/mL) with or without syringin or phillygenin (10 μg/mL, 20 μg/mL, 50 μg/mL) for 24 h [33].

Selected concentrations of 10 μg/mL, 20 μg/mL and 50 μg/mL were not toxic to macrophages (Appendix A).

### 4.4. Oil Red O Staining

To visualize the lipid deposits, cells were incubated with cholesterol (20 μg/mL) for 24 h and then incubated with syringin or phillygenin (10 μg/mL, 20 μg/mL and 50 μg/mL) for the next 24 h. Cells were fixed with 10% phosphate-buffered formalin for 10 min. After being washed in DPBS and then in 60% isopropanol, macrophages were stained with filtered Oil Red O working solution at 37 °C for 1 min without light [33]. Positive-stained cells were macrophage-derived foam cells, which were observed via microscopy (Nikon TS100F, Nikon Corporation, Tokyo, Japan) and then photographed using Image software (NIS-Elements BR 6.10.02, Melville, NY, USA).

### 4.5. Flow Cytometry Analysis

The expression of CD36 on macrophages was determined by flow cytometry FACSCalibur (BD Biosciences San Jose, CA, USA). The macrophages were incubated with syringin or phillygenin (10 μg/mL, 20 μg/mL and 50 μg/mL) for 24 h and then with cholesterol (20 μg/mL) for the next 24 h. After incubation, the macrophages were collected, suspended in Cell Staining Buffer and centrifuged at 2200 RPM for 5 min at 4 °C. Cells were pre-incubated with 5 μL of human TruStain FcX for 10 min at room temperature to block non-specific bonds for the Fc receptor. After washing with Cell Staining Buffer, macrophages were incubated on ice for 15 min with fluorescently labelled anti-CD36-PE monoclonal antibodies. PE mouse IgG1,κisotype control was used as the isotope control. The mean fluorescence intensity (MFI) in the gated cell population was measured in FL2 (10,000 cells per sample) [21]. The results were expressed as the percentage of cells expressing the CD36 receptor.

### 4.6. Western Blot Analysis

The expression of CD36, ABCA1 and PPAR-γ, as well as Nrf2, was determined by western blot. The macrophages were incubated with cholesterol (20 μg/mL) for 24 h and then with syringin or phillygenin (10 μg/mL, 20 μg/mL and 50 μg/mL) for the next 24 h. After incubation, cells were lysed in ice-cold RIPA (0.5 g/mL Tris-HCl, pH 7.4, 1.5M NaCl, 2.5% deoxycholic acid, 10% NP-40, 10 mg/mL EDTA) buffer containing phosphatase (10 μL per 1 mL of buffer) and protease (40 μL per 1 mL of buffer) inhibitors, and the resulting lysates were centrifuged at 10,000 RPM for 15 min at 4 °C. The protein concentration was quantified by a standard colorimetric test (BCA Protein Assay Kit), and 15 μL of lysate was resuspended in 5 μL 4 × Laemmli Buffer, centrifuged, vortexed briefly, boiled for 5 min at 95°C, vortexed, and frozen as aliquots at −70 °C until analysis by 12% SDS-PAGE. Protein in the amount of 40 μg was transferred to nitrocellulose filters and immunoblotted with antibodies: CD36 (1:1000), ABCA1 (1:1000), PPAR-γ (1:1000), Nrf2 (1:1000) and β-actin (1:5000). Peroxidase-conjugated AffiniPure goat anti-rabbit antibody was used as a secondary antibody at a dilution of 1:1000 (Thermo Scientific, USA). Finally, the blots were incubated with chemiluminescent substrate for the detection of HRP (Thermo Scientific, USA) for 5 min [21]. Western blot analyses were quantified using ImageJ 1.54g software after the densitometric scanning of the bands (Bio-Rad, Hercules, CA, USA).

### 4.7. Immunofluorescence Assay

To determine total cholesterol, cells were incubated with cholesterol (20 μg/mL) for 24 h and then incubated with syringin or phillygenin (10 μg/mL, 20 μg/mL and 50 μg/mL), as well as kaempferol (20 μg/mL), for the next 24 h. The supernatants were collected and centrifuged at 13,000 RPM for 1 min at 4 °C. Total cholesterol was measured using the enzyme-linked immunosorbent assay according to the protocols provided by the manufacturers. Macrophages were used to visualize lipid deposits (Section 4.4).

The intracellular secretion of HO-1 in macrophages was measured by enzyme-linked immunosorbent assay according to the protocols provided by the manufacturers. The cells were incubated with cholesterol (20 μg/mL) for 24 h and then incubated with syringin or phillygenin (10 μg/mL, 20 μg/mL and 50 μg/mL), as well as kaempferol (20 μg/mL), for the next 24 h. The supernatant was removed and macrophages were washed twice with DPBS. Cells were lysed and samples were centrifuged and stored at −70 °C until analysis.

### 4.8. Positive Control and Agonist PPAR-γ

A solution of kaempferol at a concentration of 20 μg/mL was used as a positive control [34]. Rosiglitazone (RSG) was used as a PPAR-γ agonist [35].

### 4.9. Cytotoxicity Assay

The cytotoxicity of syringin, phillygenin, kaempferol, cholesterol, and syringin, phillygenin or kaempferol with cholesterol for macrophages was measured using a Cytotoxicity Detection Kit (LDH), according to the protocol provided by the manufacturer (Appendix A).

### 4.10. Statistics

The results were expressed as a mean ± SD of the indicated number of experiments. The statistical significance of differences between means was established by ANOVA with Tukey’s post hoc test. *p* values below 0.05 were considered statistically significant. All analyses were performed using Statistica v. 13.3.

## 5. Conclusions

The results of our studies clearly indicate that syringin can limit cholesterol accumulation by macrophages by inhibiting the expression of the CD36 receptor in a PPAR-dependent pathway. Phillygenin, on the other hand, increases cholesterol efflux from macrophages by activating the ABCA1 transporter in a HO-1-dependent pathway. Syringin and phillygenin are bioactive compounds with great potential in eliminating lipid deposits in macrophages, inhibiting foam cell formation and, consequently, atherosclerotic lesions in humans.

Although we obtained some significant results, this study has limitations. It should be emphasized that the studies demonstrate the unique properties of syringin and phillygenin but do not provide complete data on their actual effects. Further research is needed, including in vivo experiments and then clinical trials.

## Figures and Tables

**Figure 1 ijms-26-06444-f001:**
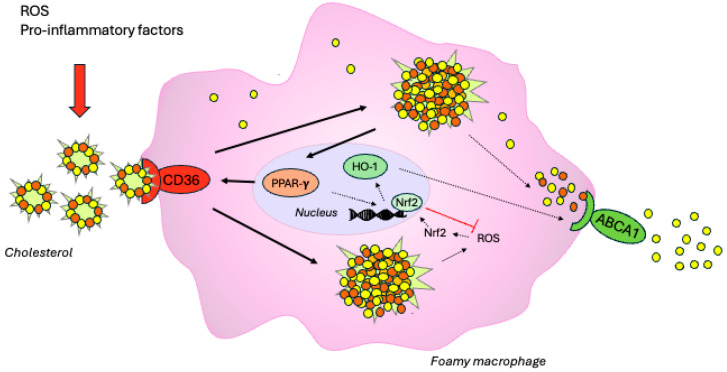
Activation of PPAR-γ in macrophages stimulates the expression of the CD36 receptor and increases lipid accumulation in cells. As a result of the unlimited uptake of cholesterol, macrophages transform into atherogenic foam cells (black arrows). Oxidative stress caused by ROS (reactive oxygen species) allows Nrf2 to enter the cell nucleus and activate the expression of genes encoding antioxidant and detoxification enzymes, such as HO-1. This enzyme can activate reverse cholesterol transport from macrophages via the ABCA1 transporter (dashed arrows).

**Figure 2 ijms-26-06444-f002:**
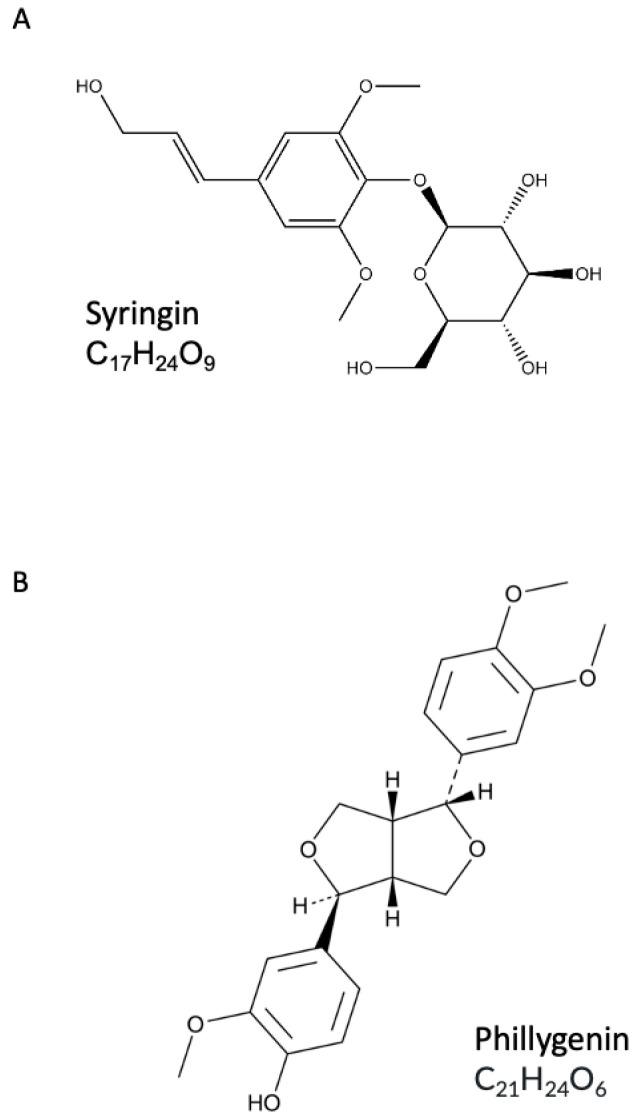
(**A**) Chemical structure of syringin. (**B**) Chemical structure of phillygenin.

**Figure 3 ijms-26-06444-f003:**
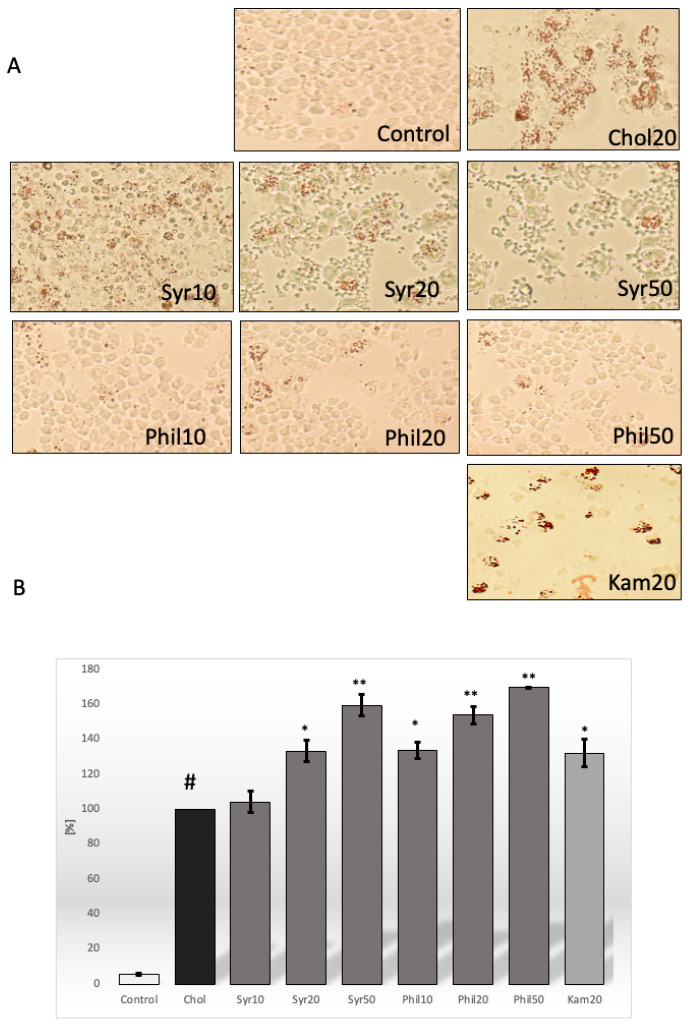
The effect of syringin or phillygenin on lipid accumulation in cholesterol-induced macrophage cells. (**A**) Oil Red O staining was used to visualize (×200 magnification) lipid deposits in macrophages (*n* = 3). (**B**) Cholesterol level in the medium of macrophages cultured with cholesterol and syringin or phillygenin compared with the medium of macrophages induced only with cholesterol (#, 100%) ± SEM (*n* = 3). Statistical significance * *p* < 0.05, ** *p* < 0.001.

**Figure 4 ijms-26-06444-f004:**
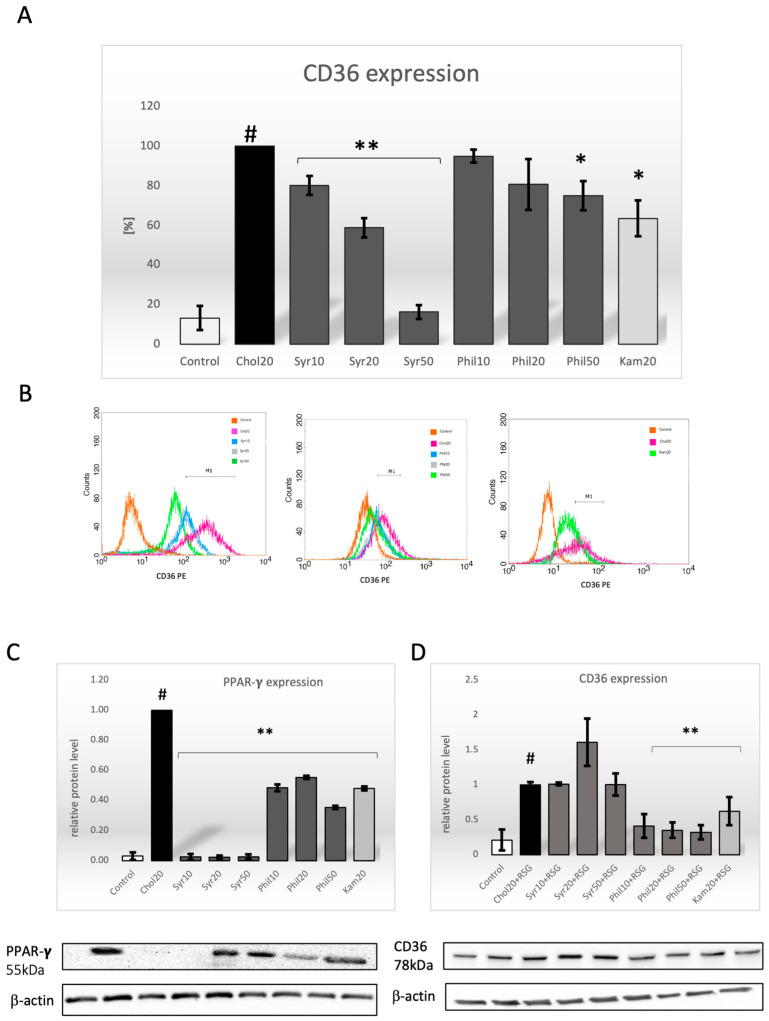
(**A**) The effect of syringin and phillygenin on CD36 expression. The flow cytometric results are presented as the percentage of cells with CD36 expression ± SEM (*n* = 3). Statistical significance * *p* < 0.05, ** *p* < 0.001 compared to cholesterol-induced macrophages (#, 100%). (**B**) The representative results are presented in histograms. CD36 expression with histogram marker M1 designating cholesterol-induced macrophages. (**C**) The effect of syringin and phillygenin on PPAR-γ protein expression. The WB results are presented as the relative PPAR-γ expression of cholesterol-stimulated macrophages (1) ± SEM *(n* = 3). Statistical significance ** *p* < 0.001 compared to Chol20 (#). (**D**) Effect of syringin and phillygenin on CD36 protein expression after preincubation with RSG. The WB results are presented as the relative CD36 expression of cholesterol-stimulated macrophages (1) after preincubation with RSG ± SEM (*n* = 3). Statistical significance ** *p* < 0.001 compared to Chol20+RSG (#).

**Figure 5 ijms-26-06444-f005:**
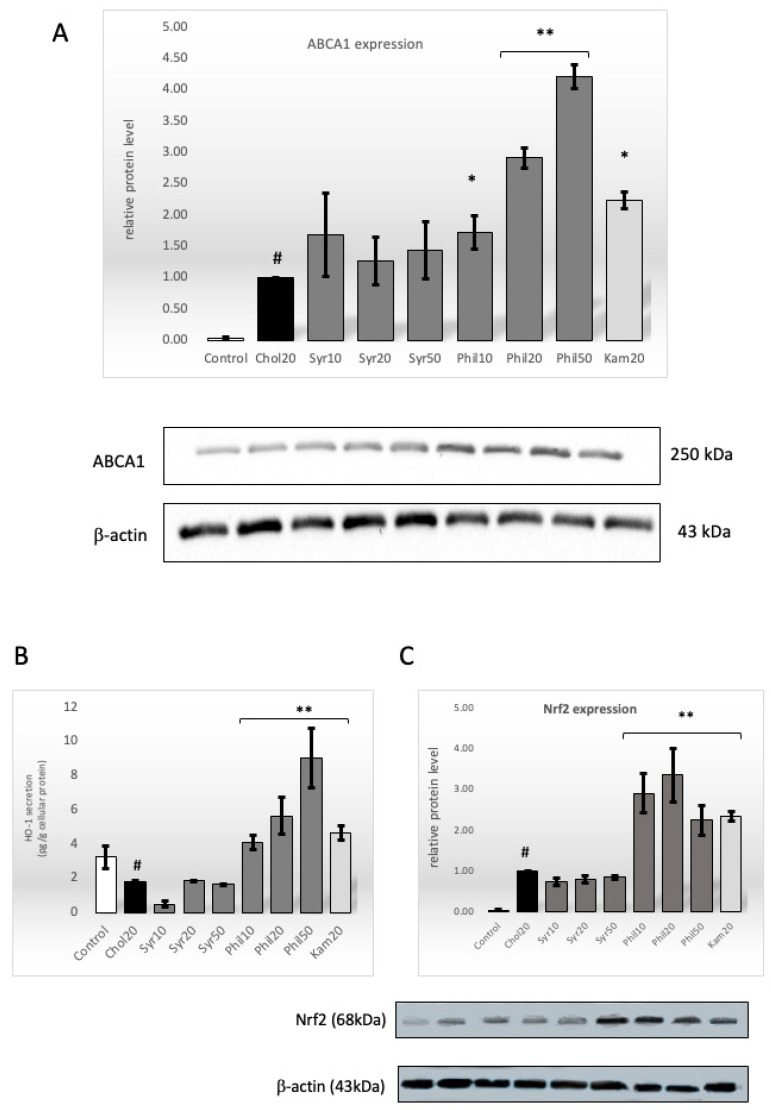
(**A**) The effect of syringin (Syr) and phillygenin (Phil) on ABCA1 expression. Quantification of protein ABCA1 expression was performed by western blot in cholesterol-induced macrophages. The results were quantified by densitometry (*n* = 3). β-actin was used as an internal control. Statistical significance ** *p* < 0.001 compared to Chol20 (#). (**B**) The intracellular secretion of HO-1 was analyzed by an ELISA test. The results are presented as pg/g of cellular protein ± SEM (*n* = 3). Statistical significance ** *p* < 0.001 compared to cholesterol-induced macrophages (#). (**C**) The protein expression of Nrf2 was analyzed by western blot in macrophages. The results were quantified by densitometry. β-actin was used as an internal control. Data are presented as the mean ± SEM for each group (*n* = 3). Statistical significance * *p* < 0.05, ** *p* < 0.001 compared to Chol20 (#).

## Data Availability

The data presented in this study are available upon request from the corresponding author.

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
