# Peer review of "Syringin and Phillygenin—Natural Compounds with a Potential Role in Preventing Lipid Deposition in Macrophages in the Context of Human Atherosclerotic Plaque"

_ijms, 2025, doi:10.3390/ijms26136444_

Round 1
Reviewer 1 Report
Comments and Suggestions for Authors
I believe that major changes are needed to the manuscript. It isn't easy to follow the results, since the authors are not citing the figures in the manuscript (for example, figures 4 and 5). I believe that IC50 values should be determined, when possible, to help compare their results. Figures and text should use the same notation. In the discussion, the authors express their results in percentages, while in the results, they report values in terms of fold over Chol20.
Author Response
I believe that major changes are needed to the manuscript. It isn't easy to follow the results, since the authors are not citing the figures in the manuscript (for example, figures 4 and 5). I believe that IC50 values should be determined, when possible, to help compare their results. Figures and text should use the same notation. In the discussion, the authors express their results in percentages, while in the results, they report values in terms of fold over Chol20.
Thank you for your valuable comments.
All figures are cited in the text and highlighted.
In the Supplementary materials section, we have added data on the cytotoxicity of the tested compounds (syringin and phillygenin), as well as cholesterol, kaempherol and rosiglitazone (Supplementary materials S1 A and B). In the Results section of the manuscript, we have included information on the non-toxicity of all the compounds used in relation to macrophages: “The IC50 value did not exceed 20% for any of the compounds at the different concentrations tested (Supplementary materials S1A). Furthermore, no toxic effect on the cells was observed when macrophages were treated with rosiglitazone at a concentration of 1 μg/mL under the same conditions (Supplementary materials S1B).”
Figures and text use the same notation.
In accordance with generally accepted principles of interpretation of scientific research results, we adopted the following rules:
- flow cytometer analysis results - percentage of cells expressing the CD36 receptor compared to cholesterol-stimulated macrophages (100%).
- western blot analysis results - relative protein level (PPAR, CD36, Nrf2) (fold) compared to cholesterol-stimulated macrophages (1).
- ELISA results - percentage of increase or decrease compared to cholesterol-stimulated macrophages.
The text has been standardized and the captions under the figures have been changed to make it clear what research technique was used.
In addition, the name of the university of one of the co-authors has been changed, as this institution became a fully-fledged university on 15 May 2025. The new name is highlighted.
Kind regards
Reviewer 2 Report
Comments and Suggestions for Authors
Dear Authors,
The manuscript "Syringin and phillygenin - natural compounds with a potential role in preventing lipid deposition in macrophages in the context of human atherosclerotic plaque" requires major revisions before it can be accepted for further publication steps, which are listed below.
Keywords - please list in alphabetical order
Fig. 1 - photo quality is poor, patterns are blurred. Please replace with a better photo.
lines 34/35 - and where is it located? You might want to add whether or not it depends on age?
line 49 - Fig. 1, but add that A
line 53 - quote #4 is too cumulative, please add separate items for each activity
line 58 - description of the next compound should be included in a separate paragraph. Please change
line 59 - incorrectly stated Fig. It is not 2, but 1B.
line 62 - quotes 8-10 please separate into individual activities. Add additional ones if necessary.
Fig. 2 - there is no reference to it in the text. It should be better described, because it takes up a large part of the Manuscript, so just mentioning it is insufficient. In addition, the diagram has a broken axis and is not very readable.
lines 74-76 should be expanded to encourage the reader to review the entire Article
In addition, the Introduction lacked a paragraph that would have recommended the compounds selected by the Authors. Why exactly these, among many others showing much broader activities, were chosen by the Authors for the study. Why not others, even belonging to the same groups? does not have this overtone of attractiveness of these compounds. Please add.
In addition, the Authors should, after the Introduction, take up the molecular/biochemical basis of atherosclerosis in a separate subsection. This subsection should include a diagram explaining the basis of its formation. It would then be clearer to the reader why the authors undertook the study of particular proteins/receptors. Please add this necessary section.
Fig. 3 - illegible, signatures blurred. The same applies to Fig. 4.
lines 79-81 - why exactly such concentrations were chosen by the Authors for the study? Please explain and justify the choice of the Authors, supporting with relevant citations.
Subsections 2.7 and 2.8 - please add graphs, possibly Figs showing the results
Please add abbreviation expansions when first used, e.g. ABCA1, PPAR, PMA..... Please review the entire Article from this point of view.
Too few references to other studies in the discussion. Have they not been performed? Perhaps the authors should consider comparing their results with similar compounds (e.g., in terms of origin, structure...). This would provide a valuable summary of the performance in this field of compounds with similar structure, containing common parts.
Subsection 4.3 - please insert superscripts and subscripts in the appropriate places. Here, the authors should also add information regarding the choice of such and not other concentrations of the tested compounds.
Subsections 4.4 and 4.5 - please provide citations for these methods. If the Authors modified them, please add such information.
4.6 - specific information is necessary, e.g. regarding the composition of the RIPA buffer with specific concentrations. These methods should be described so that it is possible to repeat them.
4.7 - the test should be briefly described. It is also necessary to give the name of the test and the company from which it was purchased.
4.9 - why this method is here, if its results have not been added to the Article. Please decide what the Authors want to include here.
In addition, the Methods lack information on the number of times each experiment was repeated. The authors count statistics, but from what?
Conclusions - should be expanded to include the Authors' proposed studies that should be done in the future to provide answers to the questions that arose from the research contained in this Manuscript. What research directions do the Authors propose to continue the research they started?
Kind regards
Author Response
The manuscript "Syringin and phillygenin - natural compounds with a potential role in preventing lipid deposition in macrophages in the context of human atherosclerotic plaque" requires major revisions before it can be accepted for further publication steps, which are listed below.
Thank you for your valuable comments.
Keywords - please list in alphabetical order
Changes were introduced. They are highlighted.
Fig. 1 - photo quality is poor, patterns are blurred. Please replace with a better photo.
We have changed the format of all figures from PDF to TIFF.
lines 34/35 - and where is it located? You might want to add whether or not it depends on age?
The word “circulating” has been added to the description of monocytes.
Information that the phenotype of macrophages depends on micro-environmental changes during atherogenesis has been added to the text.
line 49 - Fig. 1, but add that A
All figures were corrected.
line 53 - quote #4 is too cumulative, please add separate items for each activity
Individual items were listed separately.
line 58 - description of the next compound should be included in a separate paragraph. Please change
This change was introduced.
line 59 - incorrectly stated Fig. It is not 2, but 1B.
All figures were corrected.
line 62 - quotes 8-10 please separate into individual activities. Add additional ones if necessary.
Individual items were listed separately.
Fig. 2 - there is no reference to it in the text. It should be better described, because it takes up a large part of the Manuscript, so just mentioning it is insufficient. In addition, the diagram has a broken axis and is not very readable.
All figures were corrected.
The description of this figure has been added to the manuscript: “(B) Cholesterol level in the medium of macrophages cultured with cholesterol and syrignin or phillygenin compared with the medium of macrophages induced only with cholesterol (100%) ± SEM (n=3). Statistical significance *p < 0.05, **p < 0.001.”
In the Results section there are references to this figure, they are highlighted.
lines 74-76 should be expanded to encourage the reader to review the entire Article.
In addition, the Introduction lacked a paragraph that would have recommended the compounds selected by the Authors. Why exactly these, among many others showing much broader activities, were chosen by the Authors for the study. Why not others, even belonging to the same groups? does not have this overtone of attractiveness of these compounds. Please add.
The Introduction section has been expanded (changes are highlighted):
“Our previous study has also shown that common lilac is a valuable source of phytoactive compounds with anti-inflammatory properties, including syringin, which significantly inhibited TNF-α production and stimulated transforming growth factor beta (TGF-β) release in monocytes/macrophages treated with lipopolysaccharide (LPS) [13].”
“Considering the valuable properties of compounds found in plants of the Oleaceae family, including the results of our previous studies on oleacein, a secoiridoid isolated from common privet (Ligustrum vulgare), which demonstrated its anti-atherosclerotic potential [6, 21, 22], we hypothesised that syringin and phillygenin, natural compounds with high bioactivity, may also influence the prevention of cholesterol accumulation in macrophages, thereby reducing the possibility of foam cell formation. To this end, we investigated the effect of syringin and phillygenin on the regulation of CD36 receptor and ABCA1 transporter function through signaling pathways involving PPAR-γ and heme oxygenase-1 (HO-1). Understanding the effect of both compounds on selected pathogenic mechanisms of atherosclerosis may enable the development of new strategies for the prevention of early and late atherosclerotic changes.”
In addition, the Authors should, after the Introduction, take up the molecular/biochemical basis of atherosclerosis in a separate subsection. This subsection should include a diagram explaining the basis of its formation. It would then be clearer to the reader why the authors undertook the study of particular proteins/receptors. Please add this necessary section.
The Introduction section concerning atherosclerosis has been expanded (changes are highlighted):
“These receptors and transporters can be regulated via different pathways, including transcriptional modulation mediated by nuclear receptors such as peroxisome proliferator-activated receptor gamma (PPAR-γ) [1]. Activation of PPAR-γ proteins induces expression of the CD36 receptor and stimulates cholesterol accumulation in macrophages [3]. Under physiological conditions, activation of transcription factors such as erythrocyte nuclear factor 2-related factor 2 (Nrf2) initiates transcription of antioxidant genes, including heme oxidase 1 (HO-1), which protects against cholesterol accumulation [4]. HO-1 may induce the expression of the ABCA1 transporter, thereby promoting reverse cholesterol transport from macrophages [5]. Furthermore, HO-1 can suppress inflammation by promoting a shift in the macrophage phenotype from the inflammatory M1 to the anti-inflammatory M2 [6].
However, under pathophysiological conditions (chronic inflammation, oxidative factors), cholesterol efflux is insufficient, leading to unrestricted cholesterol accumulation in macrophages” (Fig. 1).
Figure 1 was also introduced.
Information explaining the reasons for researching specific proteins/receptors is also available in the Discussion section.
Fig. 3 - illegible, signatures blurred. The same applies to Fig. 4.
All figures were corrected.
lines 79-81 - why exactly such concentrations were chosen by the Authors for the study? Please explain and justify the choice of the Authors, supporting with relevant citations.
The Results section has been expanded (changes are highlighted):
“Based on our previous works [13,19], we selected concentrations of the natural compounds of 10 μg/mL, 20 μg/mL and 50 μg/mL, because these concentrations were non-toxic to cells. Moreover, higher concentrations of natural compounds (e.g. 100 μg/mL), are rarely used in scientific studies, because it is difficult to achieve such high therapeutic concentrations in the body”.
Information “Selected concentrations of 10 μg/mL, 20 μg/mL and 50 μg/mL were not toxic to macrophages (Supplementary materials S1).” was introduced to the Materials and methods section.
Subsections 2.7 and 2.8 - please add graphs, possibly Figs showing the results
These changes were introduced.
Please add abbreviation expansions when first used, e.g. ABCA1, PPAR, PMA..... Please review the entire Article from this point of view.
Expansions of abbreviations were added (they are highlighted).
Too few references to other studies in the discussion. Have they not been performed? Perhaps the authors should consider comparing their results with similar compounds (e.g., in terms of origin, structure...). This would provide a valuable summary of the performance in this field of compounds with similar structure, containing common parts.
We have not found any previous publications on the effect of syringin and phillygenin on the CD36 receptor and ABCA1 transporter.
The Discussion section has been expanded to include the effect of phenylpropanoid glycosides and lignans on the CD36 receptor and ABCA1 transporter:
“Other studies on phenylpropanoid glycosides from Tadehagi triquetrum have also demonstrated that these compounds can significantly reduce CD36 receptor expression [24].”
“These results are contrary to studies on Tadehagi triquetrum phenylpropanoid glycosides, which showed an increase in the expression of the ATP-binding cassette transporters A1 and G1 (ABCA1 and ABCG1) [24].”
“An increase in ABCA1 transporter expression has also been observed in macrophages in response to other lignans, such as arctigenin (greater burdock) [26], neolignan – honokiol (magnolia) [27], and flavonolignans – silymarin (milk thistle) [28]”
Subsection 4.3 - please insert superscripts and subscripts in the appropriate places. Here, the authors should also add information regarding the choice of such and not other concentrations of the tested compounds.
Superscript and subscript have been inserted.
We chose concentrations for compounds 10 μg/mL, 20 μg/mL and 50 μg/mL because they were not toxic to macrophages. Higher concentrations of the natural compounds, e.g. 100 μg/mL, are rarely used in scientific studies because it is difficult to achieve such high therapeutic concentrations in the body.
Information “Selected concentrations of 10 μg/mL, 20 μg/mL and 50 μg/mL were not toxic to macrophages (Supplementary materials S1).” was introduced to the Materials and methods section.
Subsections 4.4 and 4.5 - please provide citations for these methods. If the Authors modified them, please add such information.
These changes were introduced.
4.6 - specific information is necessary, e.g. regarding the composition of the RIPA buffer with specific concentrations. These methods should be described so that it is possible to repeat them.
Information concerning RIPA “(0.5 g/mL Tris-HCl, pH 7.4, 1.5M NaCl, 2.5% deoxycholic acid, 10% NP-40, 10 mg/mL EDTA) buffer containing phosphatase (10 μl per 1 ml of buffer) and protease (40 μl per 1 ml of buffer) inhibitors”, was introduced to the Materials and methods section.
4.7 - the test should be briefly described. It is also necessary to give the name of the test and the company from which it was purchased.
Test description has been added:
“QuickDetectTM Total Cholesterol Human Elisa Kit was purchased in BioVision, France. This is described in section 4.1.”
“To determine total cholesterol, cells were incubated with cholesterol (20 μg/mL) for 24 h and then incubated with syringin or phillygenin (10 μg/mL, 20 μg/mL and 50 μg/mL) as well as kaempferol (20 μg/mL) for the next 24 h. The supernatants were collected, centrifuged at 13000 RPM for 1min at 4 ◦C. Total cholesterol was measured enzyme-linked immunosorbent assay according to the protocols provided by the manufacturers. Macrophages were used to visualize lipid deposits (section 4.4).”
“The cells were incubated with cholesterol (20 μg/mL) for 24 h and then incubated with syringin or phillygenin (10 μg/mL, 20 μg/mL and 50 μg/mL) as well as kaempferol (20 μg/mL) for the next 24 h. The supernatant was removed and macrophages were washed twice with DPBS. Cells were lysed and samples were centrifuged and stored at -70oC until analysis.”
4.9 - why this method is here, if its results have not been added to the Article. Please decide what the Authors want to include here.
Cytotoxicity results were added in the Supplementary materials S1.
In addition, the Methods lack information on the number of times each experiment was repeated. The authors count statistics, but from what?
The number of times each experiment was repeated is added in the figure legends.
Conclusions - should be expanded to include the Authors' proposed studies that should be done in the future to provide answers to the questions that arose from the research contained in this Manuscript. What research directions do the Authors propose to continue the research they started?
The Conclusions section has been expanded:
“Although we obtained some significant results, this study has limitations. It should be emphasized that the studies demonstrate the unique properties of syringin and phillygenin but do not provide complete data on their actual effects. Further research is needed, including in vivo experiments and then clinical trials.”
In addition, the name of the university of one of the co-authors has been changed, as this institution became a fully-fledged university on 15 May 2025. The new name is highlighted.
Kind regards
Round 2
Reviewer 1 Report
Comments and Suggestions for Authors
The manuscript was improved after taking into account the recommendations
Reviewer 2 Report
Comments and Suggestions for Authors
Dear Authors,
Congratulations to the Authors of the revised work on the Manuscript. I am satisfied with the corrections made. I think that the Article is suitable for publication in its present form.
Kind regards